# Challenges encountered by midwives performing basic neonatal resuscitation in health facilities in Kinshasa, Democratic Republic of the Congo

Eric M. Mafuta[1], Daniel K. Ishoso[1], Carl L. Bose[2], Benjamin H. Chi[3], Patricia Gomez[4], Ingunn A. Haug[5], Helge Myklebust[5], Antoinette K. Tshefu[1], Jackie K. Patterson[2]*

1 School of Public Health, School of Medicine, University of Kinshasa, Kinshasa, Democratic Republic of Congo, 2 Department of Pediatrics, University of North Carolina at Chapel Hill, Chapel Hill, North Carolina, United States of America, 3 Department of Obstetrics and Gynecology, Department of Epidemiology, University of North Carolina at Chapel Hill, Chapel Hill, North Carolina, United States of America, 4 Jhpiego, Johns Hopkins University Affiliate, Baltimore, Massachusetts, United States of America, 5 Laerdal Medical, Stavanger, Norway

* jackie_patterson@med.unc.edu

## Abstract

Worldwide, an estimated five million children under the age of five die each year; 47% of these deaths occur during the neonatal period, and the vast majority in low- and middle-income countries. Events during labor are the cause of one quarter of neonatal deaths globally. Basic resuscitation with positive pressure ventilation reduces these deaths but is challenging to execute. To characterize barriers to implementation of basic neonatal resuscitation, we conducted a qualitative study using focus group discussions with midwives at three health facilities in Kinshasa, Democratic Republic of the Congo. We analyzed qualitative data using an inductive content approach in order to identify emergent themes and trends. Twenty-four midwives participated with a median age of 49 and over 80% with more than 10 years of clinical experience. We categorized challenges to implementing basic neonatal resuscitation into three themes with subthemes: 1) limited resources (subthemes: human resource limitations, inadequate and unprepared equipment, insufficient monitoring during labor); 2) inadequate simulated and clinical experience (subthemes: poor systems to support maintenance of skills, infrequent opportunity to resuscitate); 3) emotional burden of resuscitation (subthemes: decision-making under time pressure, tendency to stick to the routine, acute stress during resuscitation, moral distress after unsuccessful outcome). Our findings suggest that while simulation training is key, learning from clinical events may be a critical companion to address these barriers. We call for a new focus on developing and evaluating strategies that support providers in learning from every newborn resuscitation.

**Data availability statement:** All data files are available from the UNC Dataverse database https://dataverse.unc.edu/dataset.xhtml?persistentId=doi:10.15139/S3/O1RYNK.

**Funding:** JP 1R21HD103058-01 National Institute of Child Health and Human Development https://www.nichd.nih.gov/ The funder did not play any role in the study design, data collection and analysis, decision to publish, or preparation of the manuscript.

**Competing interests:** I have read the journal's policy and the authors of this manuscript have the following competing interests: JP: Received research funding from the National Institute of Child Health and Human Development, the Laerdal Foundation, the Doris Duke Charitable Foundation, the Thrasher Foundation and the Gates Foundation; she is also the recipient of a Laerdal Global Health monetary gift to support her research. CB: Received funding from the National Institutes of Health as well as travel support from the American Academy of Pediatrics and Laerdal Global Health. HM & IH: Employed by Laerdal Medical, a sister company to Laerdal Global Health. The commercial affiliation with Laerdal Medical does not alter our adherence to PLOS ONE policies on sharing data and materials.

## Introduction

Worldwide, an estimated five million children under the age of five die each year. [1] Around 47% of these deaths occur during the neonatal period (the first 28 days following birth), with approximately half of neonatal deaths happening within the first 24 hours after birth [1]. The vast majority of deaths under five occur in low- and middle-income countries [2]. Regionally, Southern Asia and Sub-Saharan Africa carry the largest burden of neonatal deaths (36% and 43%, respectively). In particular, the Democratic Republic of the Congo (DRC) is among the five countries which account for half of all newborn deaths globally [1]. Despite decrements in neonatal mortality over the past few years, the DRC still faces a high neonatal mortality rate of 26 per 1,000 live births [3].

Although healthcare provided by midwives is a core strategy for improving newborn health in low-resource settings, the midwifery profession in the DRC is challenged by poor quality pre-service training, unsupportive organizational systems and inadequate pre-conditions in the work environment [4,5]. Despites these challenges, Congolese midwives view their profession as a calling and love their work, motivating them to continue in their workplace despite the difficult work environment and low professional status [4]. Enhancing the quality of care delivered by midwives in the DRC is critical to improving maternal and newborn health outcomes.

While neonatal death is multi-factorial, intrapartum-related events account for one-fourth of neonatal deaths and present with failure to initiate and sustain breathing at birth [6]. Each year, up to 20 million neonates exposed to intrapartum-related events require assistance to breathe at birth via stimulation and/or bag-mask ventilation (BMV) [7]. The burden of intrapartum-related events is particularly high in low-resource settings where the quality of intrapartum obstetric care is poor. In these settings, emergency obstetric care and quality neonatal resuscitation are critical to reducing neonatal death.

Basic resuscitation consisting of tactile stimulation, airway positioning and clearing, and BMV reduces intrapartum-related neonatal death by 30% [8,9]. Delays in initiating resuscitation exacerbate hypoxia and can lead to neonatal morbidity and mortality.[9] In a study in Tanzania, the risk of death or prolonged admission increased by 16% for every 30-second delay in initiating ventilation up to six minutes after birth, and increased by 6% for every minute that BMV was delayed [10]. Therefore, timely resuscitation with initiation of ventilation within the first minute is an important marker of quality care. However, even with training in resuscitation, providers are challenged to execute the steps of resuscitation quickly and routinely fall short of the target of initiating BMV by one minute after birth [11]. Specifically in the DRC, we have demonstrated that even after training in resuscitation, midwives are delayed in initiating ventilation (median time to ventilation of 347 seconds after birth), and ventilation is frequently interrupted with stimulation and suctioning [12]. A decline in knowledge and skills following training has been clearly identified as a prominent barrier to quality newborn resuscitation [13,14]. However, little is known about the other barriers that contribute to poor performance of neonatal resuscitation such as emotional experiences and their influence on bedside decision-making. These barriers may

have a prominent influence on performance given prior research indicating increased stress in healthcare workers during neonatal resuscitation [15]. Furthermore, cumulative number of losses experienced is a predictor of stress, and thus stress during neonatal resuscitation in low-resource settings may be particularly high given the relatively high incidence of adverse outcomes [15]. Investigation of such barriers requires qualitative methods to illuminate the lived experiences of midwives and explore the complexity of their decision-making during urgent situations. In this study, we sought to understand barriers faced by midwives in the DRC related to implementation of basic neonatal resuscitation.

## Materials and methods

### Setting

This study took place in three Catholic health facilities situated in three different health zones in Kinshasa, the capital city of the DRC. The approximate number of annual births at these facilities ranged from 1,300–3,900. Approximately 3% of neonates received BMV at birth at these facilities, and the neonatal mortality rate was approximately 13 per 1000 live births. All of the facilities practiced basic resuscitation; none of the three facilities used electronic fetal heart monitoring during labor, and none offered Cesarean section.

### Study design and procedures

We conducted focus group discussions (FGDs) at three health facilities that were purposively selected due to their relatively high number of deliveries and their recent participation in research studies focused on neonatal resuscitation led by members of our team.

Midwives were eligible to participate if they provided newborn resuscitation at birth as part of their regular employment. With the assistance of the head nurse midwives, we purposively sampled a diverse group of midwives representing a range of ages, years of clinical experience and location of employment in order to capture a variety of perspectives. Midwives were assigned a participant identification number that included their FGD number as a preface (i.e., FGD#-M#).

Two male team members (EM and DI) with prior qualitative data collection experience conducted FGDs in February 2021. EM has an MD, MPH and a PhD in global health and was an associate professor at the Kinshasa School of Public Health; DI has an MD and was a PhD student in maternal and child health. While EM did not have a prior relationship with the participants, many of the participants knew DI as he had recently coordinated a clinical trial on newborn resuscitation in those sites. The FGDs were held in a quiet, private setting at the respective health facility and lasted on average one hour. To minimize facilitator bias, the facilitators used an FGD guide to structure the discussion, engaged in active listening, and asked open-ended questions. In each FGD, participants were invited to discuss their experiences resuscitating newborns, with a focus on challenges they encountered when carrying out newborn resuscitation. Specific themes explored included the following: What are the challenges to providing newborn resuscitation in your health facility? What are the challenges to correctly perform the steps of newborn resuscitation? What happens when all the appropriate steps are not taken during resuscitation, or when the baby dies despite everyone's best efforts? To minimize groupthink bias, the facilitators were neutral in their responses, encouraged individual opinions, and discouraged side conversations among participants.

FGDs were conducted in both Lingala and French. We audio-recorded each FGD with the consent of the participants and summarized each FGD in a brief report soon after. The reports were then orally reviewed with participants to validate the findings immediately following the FGD. Our team members also debriefed together after each FGD, noting themes and recording their impressions of the findings and procedures in field notes.

### Data processing and analysis

We transcribed recorded FGDs in Lingala and/or French as spoken during the FGD. We then translated comments in Lingala to French and then, where needed, French to English. Transcripts were back-translated and reviewed by bilingual

team members to validate accuracy. Using Atlas-ti 9 (Scientific Software development, Berlin), we analyzed transcripts following an inductive content approach in order to identify emergent themes and trends in the data [16–18]. First, French transcripts were read and re-read by two research team members to familiarize themselves with the content of the transcripts. For this step, we used a voice-centered relational method with reflexive elements built in that included reading the interview text three or more times. One of the readings involved using a worksheet technique to lay out the respondent's words in one column and react to and interpret the text in an adjacent column [19]. In this 'reader-response' technique, the researcher puts himself, his background, history and experiences in relation to the respondent, reading the narrative on his own terms and listening for how he is responding emotionally and intellectually to this person. This allows researchers to examine how and where some of their assumptions and views might affect their interpretation of the respondent's words, or how they later write about the person. For example, since both facilitators were physicians with prior experience providing newborn resuscitation, they were aware of the tendency for midwives to suction all babies, including those who were breathing. This prior experience influenced the way they coded quotes related to suctioning, ultimately assigning them a category of 'tendency to stick to the routine.' In a second step, both readers independently developed a coding sheet, assigning codes that emerged from reading. With each subsequent FGD, they assessed consistency in the data, meaning saturation and code saturation to determine if thematic saturation had been reached. In a third step, the two readers then discussed their coding sheets and together created sub-categories and categories by consensus. Subsequently, we labeled participant individual comments, counted them to provide figures and orders, and then organized them into labels, sub-categories, and categories (in ascending order). The team used notes from the meetings to corroborate their categorization schema [20] and refined as needed. While participants were not approached to provide feedback on the findings, analytical outputs were refined through discussion with other research team members from different backgrounds and cultures.

### Ethical approval

This study was approved by both the Kinshasa School of Public Health Institutional Review Board (IRB; KSPH IRB# ESP/CE/18/2021) and the University of North Carolina at Chapel Hill IRB (UNC IRB #20–3414). We approached all potential participants face-to-face for written informed consent prior to participation. To preserve anonymity and confidentiality, all identifying information was excluded from transcripts, and quotations for reporting were attributed to specific participants using a participant ID.

## Results

We conducted FGDs with a total of 24 midwives (100% of those approached consented) with a median age of 49. Over 80% had more than 10 years of clinical experience (Table 1). Only the participants and researchers were present for the FGDs.

The challenges faced by midwives in implementing basic neonatal resuscitation were organized into three themes with subthemes per Table 2.

### Limited resources

Midwives identified understaffing, lack of prepared equipment and inadequate monitoring of the fetus during labor as challenges to implementing basic neonatal resuscitation.

**Subtheme: Human resource limitations.** Several midwives mentioned challenges related to insufficient number and quality of trained personnel. Themes cited by midwives included working alone without support (n = 11), limited number of staff when a resuscitation case occurred (n = 7), lack of pediatricians for complex cases (n = 2), and the need to manage several cases at once (n = 1). Midwives recognized the importance of having more than one person providing care during a resuscitation:

**Table 1. Demographics.**

| Demographic | Midwives (n = 24) n (%) |
|---|---|
| Age, median (IQR) | 49 (40, 57) |
| Education | |
| Nursing school (5 years post primary) | 3 (12.5) |
| Secondary school (6 years post-primary) | 9 (37.5) |
| Licensed (up to 8 years post-primary) | 12 (50.0) |
| Years of clinical experience | |
| 10 or less years | 4 (16.7) |
| More than 10 years | 20 (83.3) |

*"In order for our resuscitation to go as it should with respect for all stages, although we have all the necessary equipment, although we have qualified staff, we must be at least two during the delivery. The contribution of two people is better than that of one person. The day shift is much larger than the night shift. At night, there are generally three people, while during the day at least seven people." (FGD2-M5)*

Midwives also described the challenge of resuscitating a newborn when a maternal delivery complication occurs, as midwives have to take care of the mother while also caring for the newborn (n = 2; Table 2).

**Subtheme: Inadequate and unprepared equipment.** A lack of functioning equipment prevented midwives from carrying out an efficient resuscitation (n = 6). For example, intermittent lack of electricity for powering light and radiant heat on the warmer bed prevented them from optimally warming the baby.

Midwives mentioned insufficient preparation of equipment for childbirth as a challenge of efficient resuscitation (n = 4). They noted that equipment may not be prepared in advance when a woman is admitted with full dilation and delivers quickly, or when midwives are busy taking care of other tasks and do not anticipate an impending birth. The midwives emphasized when there is a need for resuscitation in a delivery room which is not prepared beforehand, case management becomes very difficult (Table 2).

**Subtheme: Insufficient monitoring during labor.** Midwives described insufficient monitoring during labor (n = 6) including neglecting to evaluate the fetal heart rate (FHR), not evaluating labor progress including cervical dilatation, and failing to detect symptoms such as maternal fever that can lead to fetal distress. This inadequate monitoring sometimes led to an unexpectedly depressed newborn who was more difficult to resuscitate (Table 2). Consistent monitoring during labor could assist in risk stratification and allocation of resources for high-risk newborn cases; it could also prompt intrapartum interventions to reduce fetal distress and ultimately improve the newborn's condition at birth.

**Inadequate simulated and clinical experience.** Midwives identified unstructured systems to assist them in maintaining skills and lack of frequent clinical use of resuscitation skills as significant challenges to implementation of basic neonatal resuscitation.

**Subtheme: Poor systems to support maintenance of skills.** Midwives articulated many themes related to poor systems to support maintenance of skills, including the inadequate re-training of staff (n = 3), lack of skills practice (n = 3) and the absence of action plans in the delivery room (n = 3). The need for on-going training was linked to the next subtheme of infrequent opportunity to resuscitate; systems to maintain skills that are infrequently used in clinical care are key to maintaining provider competency in and comfort with resuscitation.

**Subtheme: Infrequent opportunity to resuscitate.** The insufficient number of clinical resuscitations (n = 4) was also identified as an important challenge to implementing resuscitation. For most, it had been a long time since they carried

**Table 2. Summary of themes with representative quotes.**

| Theme | Representative quote |
|---|---|
| **Limited resources** | |
| Human resource limitations | *"We take immediate care of the delivery of the mother and her baby. The difficulty of taking care of two at the same time arises when the mother has a hemorrhage, then we do not know how to share. However, once we had prepared our resuscitation table well, we could already finish caring for the child during the period of latency before the placental abruption." (FGD2-M6)* |
| Inadequate and unprepared equipment | *"These situations occur when the delivery table was not prepared beforehand. We know that we must prepare our table well and place all the resuscitation equipment, not knowing what could happen during a childbirth. This is when another childbirth suddenly arrives, the woman in full dilation, when the room has not been prepared in advance, when midwives received a woman who had come in imminent labor with the bulging water bag, since she had not brought the needed materials for carrying out childbirth." (FGD2-M3)* |
| Insufficient monitoring during labor | *"When I received a woman in labor with normal fetal heart rate and then after delivery, we lose this baby in resuscitation, it is difficult for us to forget such an unfortunate experience. Especially after putting so much effort into resuscitating him/her and ultimately the baby dies. As an example, we had come on night shift but we had not evaluated the woman in labor. We were called in that the woman had presented a fever which we also objectified, that is when we evaluated the woman and we realized that the fetus had already entered into fetal distress." (FGD3-M4)* |
| **Inadequate simulated and clinical experience** | |
| Poor systems to support maintenance of skills | *"Most often, it is the non-retraining of resuscitation training. If the person only uses their previous knowledge and is not up to date on the steps and standards to be followed, they will have a hard time applying as they should. While the one that has been retrained will follow these different steps to the letter." (FGD2-M1)* |
| Infrequent opportunity to resuscitate | *"Maybe it's because the cases of resuscitation are not so frequent. When we just take the training if the cases are frequent, we put it in practice. There the protocol is still in mind and we practice…" (FGD1-M6)* |
| **Emotional burden of resuscitation** | |
| Decision-making under time pressure | *"We take too long to dry the child; we forget that it is an emergency and the importance of the Golden Minute. The child did not cry but more time is spent drying him/her instead of rushing for emergency resuscitation." (FGD2-M4)* |
| Tendency to stick to the routine | *"The other problem is that when the newborn is not breathing, instead of starting ventilation directly, more time is spent suctioning him/her. Suctioning that takes longer than necessary, based on the notion that ventilating a newborn with secretions is ineffective. I think we spend more time on suctioning than we need to." (FGD1-M3)* |
| Acute stress during resuscitation | *"Sometimes the stress secondary to the progress of the resuscitation can cause us to lose control of the situation, which could explain the fact that we can skip certain stages of the resuscitation. We are panicked and we are afraid that the baby may die." (FGD3-M4)* |
| Moral distress after unsuccessful outcome | *"We are often saddened by the loss of a baby that it took time and energy to save during resuscitation. I often wonder, what should I do that I didn't? We feel a certain guilt knowing that we are dedicated to helping pregnant women give birth to healthy babies, and that we also have the necessary equipment to ensure a good resuscitation of these babies, but that in the end, we can lose the baby in question despite our best efforts. Even when you get home, you think back to the situation you experienced in the hospital." (FGD2-M5)* |

out a resuscitation (Table 2). Several participants stated mastering resuscitation requires frequent practice; this may be because resuscitation involves frequent assessment of the neonate, execution of complex skills and respecting the appropriate order of steps. Simulation practice can help mitigate infrequent opportunities to resuscitate. However, the nuanced assessment of an infant during the fetal to neonatal transition and the need to master BMV on varied body types precludes simulation from serving as a replacement for bedside clinical practice.

## Emotional burden of resuscitation

Midwives identified several challenges related to the emotional burden of resuscitation including making decisions under urgent circumstances, resorting to habitual actions, acute stress when faced with a non-breathing baby, and moral distress following adverse outcomes.

**Subtheme: Decision-making under time pressure.** When discussing challenges to execute the steps of resuscitation, midwives recognized the importance of keeping track of the time elapsed since birth. They placed particular emphasis on the importance of the Golden Minute (n = 7; Table 2). Midwives asserted that time management is easier when a colleague is present to help with the resuscitation (n = 3):

*"The message is not to skip the steps of resuscitation. For example, when there are two midwives during resuscitation and I spend too much time drying the child, my companion can remind me to speed things up and take the next steps to manage the Golden Minute well and not waste this precious time unnecessarily. It is the effective management of this first minute which is the key to the success of the resuscitation of the newborn; if we do not manage this period well, we put the life of the newborn in danger."* (FGD2-M3)

**Subtheme: Tendency to stick to the routine.** Another challenge highlighted by midwives was poor adherence to resuscitation guidelines; instead, they often stuck to what was routine. While most midwives felt that they benefitted from resuscitation training, they frequently skipped resuscitation steps learned in training:

*"We also tend to fall into a routine, despite training and retraining. The tendency to stick to the routine… and not follow the steps to the letter is a real challenge."* (FGD2-M4)

In particular, midwives highlighted the tendency to stick to routine procedures such as suctioning (Table 2). The tendency to suction was linked to the subtheme of acute stress during resuscitation; in this context, reverting to suction may reflect a reliance on ingrained habits as midwives tend to suction not only non-breathing newborns but also breathing newborns. Reverting to suction may also reflect lack of comfort with and confidence in BMV and thus hesitation to initiate this therapy.

**Subtheme: Acute stress during resuscitation.** Stress, panic, and agitation associated with resuscitation were also mentioned by midwives as a challenge for carrying out effective resuscitation (n = 5). Sometimes the birth of a newborn that necessitates resuscitation, or the inappropriate progress of the resuscitation, could cause them to panic (Table 2). Midwives described that when emotions are high, mistakes could be made:

*"Agitation and fear. When you see that the baby is not crying you panic. Instead of respecting all the stages, you start with the third and then the first. You act in haste."* (FGD1-M4)

Stress was also linked to subthemes of human resource limitations (i.e., resuscitating alone) and infrequent opportunity to resuscitate.

**Subtheme: Moral distress after unsuccessful outcome.** In reflecting on the challenge of resuscitation, many midwives described scenarios that demonstrated grappling with guilt (n = 5; Table 2). One midwife described how she reacts to a resuscitation that does not result in a crying baby:

*"I often sit next to the newborn watching him for at least 15 minutes, hoping he can breathe again after all efforts to resuscitate. I often regret the time, especially if we have lost the Golden Minute, the period during which resuscitation must be most intensive."* (FGD2-M6)

This burden of moral distress may be mitigated by a facility culture of teamwork and improvement.

## Discussion

The present study described common challenges encountered by midwives in the DRC that prevent them from effectively carrying out neonatal resuscitation. Some of our findings have been commonly reported in the literature, including human

resource limitations, inadequate and unprepared equipment, poor systems to support maintenance of skills and infrequent opportunity to resuscitate. Our subtheme of insufficient monitoring during labor highlights a challenge common to many low-resource settings. Of particular interest were the subthemes around the emotional burden of resuscitation, including decision-making under time pressure, tendency to stick to the routine, acute stress during resuscitation, and moral distress after unsuccessful outcome.

Under the theme of limited resources, the subtheme of inadequate staffing and equipment is a common issue relevant to quality care in low-resource settings. Midwives in our study emphasized the importance of not resuscitating alone, even though human resource limitations made that a common occurrence. The perceived importance of team-based resuscitations is consistent with prior studies in Tanzania, where midwives reported benefits of performing resuscitation with colleagues (e.g. exchange of ideas, complementary skills) [21,22]. Novel strategies to support midwives under limited staffing conditions such as automated real-time guidance or telehealth could be evaluated to address this barrier. Availability of resuscitation equipment has also been commonly cited as a barrier to quality newborn resuscitation in low-resource settings, including in the DRC [4,23], and is positively associated with neonatal survival [9,24]. Eliminating equipment-related barriers is a key step to improving resuscitation care, and should include systems-based solutions that facilitate consistent preparation of equipment prior to every birth.

Importantly, in the theme of inadequate simulated and clinical experience, midwives in our study perceived that the frequency with which they are able to practice their skills is a critical barrier to effective performance. Most of the literature around maintenance of newborn resuscitation skills has focused on the importance of simulation training, whether in the form of refresher trainings or low-dose high-frequency skills practice with a manikin [25]. Midwives in our study felt that clinical practice is equally important, and that poor systems to support maintenance of skills and infrequent opportunities to resuscitate were challenges to effective implementation. In prior research, we estimated the annual number of BMV occurrences for a given skilled birth attendant in a variety of low-resource settings, with numbers ranging from one to 23 per year [26]. With limited opportunities to practice clinically, maximizing learning during clinical events may be particularly important. For example, a recent study on the champion midwife program in Tanzania embedded hands-on coaching of BMV during clinical newborn resuscitations following HBB training, and found improved self-efficacy [21].

The influence of emotion on resuscitation performance, and the subtheme of a tendency to stick to the routine, was illustrated by the use of suctioning during resuscitation. Frequent suctioning was highlighted by midwives as a common deviation from resuscitation guidelines. One hypothesis for the tendency to suction is that midwives do not understand the clinical significance of the initial steps of resuscitation, and particularly BMV [24]. Alternatively, it may be that midwives are comfortable with routine practices and less comfortable with complex practices such as BMV; thus, they hesitate to initiate BMV, hoping that suctioning will suffice. Strategies to de-implement suctioning may be key to improve both timely and effective BMV.

Also under the theme of emotional burden of resuscitation were subthemes of acute stress and moral distress. A qualitative study with midwives from Haydom Lutheran Hospital in Tanzania identified that the stress of ventilating a non-breathing baby produced anxiety and fear, often leading to poor resuscitation performance [27]. This reaction has also been reported in high-resource settings; midwives in Norway claimed newborn resuscitation as one of the most frightening situations they can experience, with fear of perceptions of incompetence, blame or reproachment [22]. Stress and arousal in response to a resuscitation may not only impede performance, but also hamper learning during clinical care.[21] Simulation practice is a common strategy to enhance preparation for these stressful scenarios; however, additional strategies that reduce stress in the moment may be needed to enhance performance. Finally, we were struck by the theme of moral distress, and particularly guilt, in our conversations with midwives. Feelings of culpability may reflect low psychological safety and negatively affect learning from these cases. Strategies that support a culture of improvement at the facility level, rather than blame and shame, may further enhance learning from clinical resuscitations.

The challenges noted in this study may be framed by the Cognitive Load Theory [28]. Cognitive load (the total amount of mental effort used in working memory) is made up of intrinsic, germane and extraneous components. The inherent difficulty of resuscitation, and particularly the skill of BMV, contributes to a high intrinsic load; this, combined with the stress of resuscitating a non-breathing newborn (extraneous load), leaves limited room for learning (germane load). Strategies that mitigate stress during resuscitation (thereby reducing extraneous load) and that ensure efficient learning (thereby optimizing germane load), may be key to addressing challenges to implementation of basic resuscitation.

Many of the challenges noted in our study may be influenced by organizational culture, peer support, supervisory practices and leadership. Facility culture that promotes an environment of improvement may mitigate feelings of moral distress with resuscitation; peer support may similarly mitigate moral distress. Reflective supervision, including on-the-job coaching and team-based debriefings, may help to address both technical and emotional challenges encountered with resuscitation. Debriefing may be implemented through structured post-resuscitation debriefs or peer reflective sessions; mobile health tools may bolster such strategies in settings with limited experience debriefing. Leadership and managerial training for supervisors of maternal and newborn health workers in health facilities may also mitigate challenges; in Morogoro, Tanzania, training for supervisors resulted in enhanced teamwork, job satisfaction, productivity and improved care [29]. Investment in leadership may be an overarching strategy to empower ownership of challenges and improve implementation of basic newborn resuscitation.

Despite our many strengths, we note limitations to our study as well. Our sample size was small and most of the respondents were of similar age. Our findings may be susceptible to social desirability bias due to the use of FGDs, and to recall bias due to the retrospective nature of our data collection. Analysis of the data was conducted primarily by two male researchers with prior experience with labor, delivery and neonatal resuscitation during medical training; both also had previous experience in maternal health research. While their background and experiences may have influenced the interpretation of the data, we attempted to minimize this bias with our reflexive approach to analysis. Although our sample was drawn from high-performing urban facilities, we believe the challenges identified—particularly the emotional burden of resuscitation—are likely transferable to similar low-resource maternity settings. Further research is needed to explore these dynamics in rural or under-resourced contexts.

## Conclusions

Through this qualitative study, we identified challenges to implementation of basic resuscitation from the perspective of frontline midwives. The emotional burden of resuscitation, including decision-making under time pressure, tendency to stick to the routine, acute stress during resuscitation and moral distress after unsuccessful outcomes, illustrates the challenge of translating training into practice. While simulation training is key, learning from clinical events may be a critical companion to address these barriers. The insights from this study can inform the design of future interventions to improve newborn resuscitation care, including such implementation strategies as coaching, clinical debriefing, and supportive supervision. We are currently evaluating mobile health-supported real-time guidance and debriefing as strategies to learn from clinical resuscitations in a trial in the DRC [30]. Our findings in this qualitative study call for a new focus on developing and evaluating strategies that support providers in learning from every newborn resuscitation.

## Supporting information

**S1 File. Inclusivity in global research questionnaire.**
(DOCX)

**S2 File. Basic Resuscitation Challenges COREQ_Checklist.**
(PDF)

## Author contributions

**Conceptualization:** Eric M. Mafuta, Daniel K. Ishoso, Carl L. Bose, Benjamin H. Chi, Patricia Gomez, Ingunn A. Haug, Helge Myklebust, Jackie K. Patterson.

**Data curation:** Eric M. Mafuta, Daniel K. Ishoso.

**Formal analysis:** Eric M. Mafuta, Daniel K. Ishoso, Jackie K. Patterson.

**Investigation:** Eric M. Mafuta, Daniel K. Ishoso.

**Methodology:** Eric M. Mafuta, Daniel K. Ishoso, Jackie K. Patterson.

**Project administration:** Jackie K. Patterson.

**Supervision:** Carl L. Bose, Antoinette K. Tshefu.

**Validation:** Jackie K. Patterson.

**Writing – original draft:** Eric M. Mafuta, Jackie K. Patterson.

**Writing – review & editing:** Eric M. Mafuta, Daniel K. Ishoso, Carl L. Bose, Benjamin H. Chi, Patricia Gomez, Ingunn A. Haug, Helge Myklebust, Antoinette K. Tshefu, Jackie K. Patterson.

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
