## [Decision Letter · Decision Letter 0]

2 Apr 2025

Dear Dr. Patterson,

Thank you for submitting your manuscript to PLOS ONE. After careful consideration, we feel that it has merit but does not fully meet PLOS ONE’s publication criteria as it currently stands. Therefore, we invite you to submit a revised version of the manuscript that addresses the points raised during the review process.

We look forward to receiving your revised manuscript.

Kind regards,

Dawit Getachew Gebeyehu, MPH

Academic Editor

PLOS ONE

Journal Requirements:

“I have read the journal's policy and the authors of this manuscript have the following competing interests:

JP: Received research funding from the National Institute of Child Health and Human Development, the Laerdal Foundation, the Doris Duke Charitable Foundation, the Thrasher Foundation and the Gates Foundation; she is also the recipient of a Laerdal Global Health monetary gift to support her research.

 CB: Received funding from the National Institutes of Health as well as travel support from the American Academy of Pediatrics and Laerdal Global Health.

HM: Employed by Laerdal Medical, a sister company to Laerdal Global Health.

IH: Employed by Laerdal Medical, a sister company to Laerdal Global Health.”

We note that one or more of the authors are employed by a commercial company: Laerdal Medical

Additional Editor Comments:

While revising this manuscript please make sure PLOS one manuscript submission guideline was followed

Reviewers' comments:

Reviewer's Responses to Questions

**Comments to the Author**

1. Is the manuscript technically sound, and do the data support the conclusions?

Reviewer #1: Partly

Reviewer #2: Partly

2. Has the statistical analysis been performed appropriately and rigorously?

Reviewer #1: N/A

Reviewer #2: N/A

3. Have the authors made all data underlying the findings in their manuscript fully available?

Reviewer #1: No

Reviewer #2: No

4. Is the manuscript presented in an intelligible fashion and written in standard English?

Reviewer #1: Yes

Reviewer #2: No

Reviewer #1: Dear Authors,

Thank you for the opportunity to review your manuscript. Your study presents a well-structured qualitative analysis of the challenges midwives face in implementing neonatal resuscitation in the Democratic Republic of the Congo (DRC). It effectively highlights key barriers, including human resource limitations, inadequate equipment, stress, and emotional burden. Below are some comments that I hope will be helpful in refining your manuscript for publication.

1. Introduction

• The introduction outlines neonatal mortality statistics and the importance of timely resuscitation, but it does not explicitly state the gap in existing knowledge that this study seeks to fill.

• Please provide the reader with further details about basic resuscitation in DRC.

2. Methods

• The context in which the study is taking place is missing. Please include a paragraph about your setting in which the study takes place. What care level where the facilities at? Number of births? Neonatal deaths, Resuscitations, etc.

• The description of the focus group discussions (FGDs) is detailed, but some additional clarifications would improve transparency.

o Were there any efforts to minimize groupthink bias in FGDs?

o Did the facilitators have prior relationships with participants that could have influenced responses?

• As a reader, I am not entirely clear on the type of analysis you have conducted. The analysis procedure refers to both content and thematic analysis, making it difficult to understand your analytical process and how your themes emerged. For example, you reference Elo and Kyngäs, but based on my understanding, your analysis does not align with theirs. Please revisit your references and provide a clearer explanation of how your content analysis was conducted. This is particularly important since you have counted your themes, which is not mentioned in the analysis section.

3. Results

• The manuscript often states the number of midwives mentioning a particular theme (e.g., "n=6"), but this is inconsistent. Some sections include this, while others do not. Consider whether the quantification is required, and if so, be consistence

• The results would benefit from a more in-depth analysis.

4. Discussion

• The discussion relates findings to previous research, but it lacks a discussion in relation to research on midwives in DRC (simulation, education, work environment).

• The limitations section is well written but could be expanded to discuss:

o Possible social desirability bias in FGDs.

o The generalizability of findings beyond Kinshasa, given urban-rural disparities in healthcare resources.

Reference tips:

o Berg et al (2022) Implementation of a three-pillar training intervention to improve maternal and neonatal healthcare in the Democratic Republic Of Congo: a process evaluation study in an urban health zone

o Bogren et al (2020) Midwives’ challenges and factors that motivate them to remain in their workplace in the Democratic Republic of Congo—an interview study

o Bogren et al (2021) Barriers to delivering quality midwifery education programmes in the Democratic Republic of Congo — An interview study with educators and clinical preceptors - ScienceDirect

o Baba et al (2020) 'Being a midwife is being prepared to help women in very difficult conditions': midwives' experiences of working in the rural and fragile settings of Ituri Province, Democratic Republic of Congo - PubMed

Reviewer #2: This manuscript addresses a critically important issue in global maternal and newborn health by exploring the challenges encountered by midwives in implementing basic neonatal resuscitation in the Democratic Republic of the Congo (DRC). The qualitative design is appropriate for the research question, and the study provides rich, meaningful insights into both structural and psychological barriers to quality neonatal care in low-resource settings. It is a valuable contribution to the field and presents data that are largely consistent with the conclusions drawn.

However, several key areas require attention and improvement before the manuscript can meet the methodological and editorial standards expected by PLOS ONE. My detailed comments are as follows:

1. Abstract

Please clarify in the abstract that this is a qualitative study employing focus group discussions and inductive content analysis. At present, the methodology is described in general terms, which may hinder readers from immediately identifying the study design. I recommend adding the phrase:

"We conducted a qualitative study using focus group discussions and inductive content analysis..."

2. Introduction

a) Consider explicitly articulating the gap in the literature, particularly concerning midwives’ emotional experiences and bedside decision-making. This would position your study more clearly as a novel contribution. You may also cite relevant literature on emotional labor or the psychological burden associated with resuscitation to enhance theoretical grounding.

b) While the rationale for examining implementation barriers is sound, the justification for using a qualitative design should be expanded. Explain why qualitative methods were necessary to capture midwives’ experiences, and how they provide unique insights beyond what quantitative studies offer.

3. Materials and Methods

a) Study Design and Sampling

Expand on interviewer characteristics (e.g., gender, professional background, training) and any steps taken to minimize interviewer bias. These are key elements of the COREQ checklist and contribute to the transparency and trustworthiness of the study.

b) Procedure and Language

Clarify how translation fidelity was ensured in the process from Lingala to French and then English. This is essential for data trustworthiness. Please state whether transcripts were back-translated or reviewed by bilingual team members to validate accuracy.

c) Data Analysis

Provide specific examples of how reflexivity and the worksheet technique influenced coding or thematic interpretation. Currently, the reflexivity component appears superficial. Additionally:

How many researchers were involved in coding?

Was intercoder agreement assessed or reached by consensus?

Provide one or two concrete instances of how reflexive insights shaped the interpretation.

4. Ethical Considerations

Include a brief statement on how anonymity and confidentiality were maintained throughout the process—particularly during transcription, translation, and reporting of participant quotations. This will strengthen the ethical transparency of the manuscript.

5. Results

a) Some themes appear conceptually overlapping or insufficiently differentiated—for example, "stress," "time management," and "managing guilt" could be grouped under a broader domain such as Emotional Burden of Resuscitation. Consider reorganizing themes for greater analytic coherence. A possible structure:

Main Theme: Emotional Burden of Resuscitation

Subtheme 1: Acute stress during resuscitation

Subtheme 2: Time pressure and decision-making

Subtheme 3: Moral distress and guilt after unsuccessful outcomes

A thematic table summarizing categories, subthemes, and representative quotes would enhance clarity and accessibility.

b) Several themes are described only superficially (e.g., "tendency to stick to the routine"). Consider going beyond surface-level description to explore underlying mechanisms. For example: Why do midwives revert to suctioning? Is it due to uncertainty, ingrained habits, fear of failure, or a lack of confidence in BMV?

c) Quotation labels are inconsistently formatted (e.g., “M3,” “M4_1”) and often lack contextual information (e.g., focus group number or facility type). Use a standardized format such as “FGD1-M3,” and briefly explain the labeling system in the methods section to improve auditability.

d) There is no mention of whether thematic saturation was achieved. Please briefly describe whether and how saturation was assessed across the three FGDs.

6. Discussion

a) Reorganize the discussion section by clustering key findings under broader conceptual domains to enhance interpretive clarity. Suggested structure:

Structural/Systemic Barriers (e.g., human resources, equipment, training translation);

Psychological and Emotional Burdens (e.g., stress, guilt, time pressure);

Behavioral Patterns and Learning Gaps (e.g., sticking to routine, suctioning overuse)

b) While the discussion references literature from similar settings, it lacks deeper theoretical framing that could contextualize the observed behaviors. Consider integrating frameworks such as Emotional Labor Theory or Cognitive Load Theory, which would enhance the depth of analysis and support a more interpretive, theory-informed approach to the qualitative findings.

c) Expand on the role of organizational culture, peer support, and supervisory practices in shaping midwives’ resuscitation behaviors. Discuss how reflective supervision, on-the-job coaching, and team-based debriefings could address both technical and emotional challenges.

d) The recommendation for real-time debriefing and learning from clinical events is highly relevant but underdeveloped. Provide specific examples of how such strategies could be implemented (e.g., structured post-resuscitation debriefs, peer reflective sessions, mHealth tools). You may reference Helping Babies Breathe (HBB) as a concrete model that operationalizes bedside learning in low-resource settings (Bang et al., 2016).

Reference: Bang, A., Patel, A., Bellad, R., Gisore, P., Goudar, S. S., Esamai, F., ... & Hibberd, P. L. (2016). Helping Babies Breathe (HBB) training: What happens to knowledge and skills over time?. BMC pregnancy and childbirth, 16, 1-12.

e) Line 339 suggests that findings may be generalizable to other settings. This may overstate the transferability of qualitative research. Consider revising the sentence to:

“Although our sample was drawn from high-performing urban facilities, we believe the challenges identified—particularly emotional and behavioral barriers—are likely transferable to similar low-resource maternity settings. Further research is needed to explore these dynamics in rural or under-resourced contexts.”

7. Conclusion

Strengthen the conclusion by explicitly linking your findings to future implementation strategies. For example, emphasize how insights from this study can inform the design of interventions such as: Team-based care models,

Simulation-enhanced coaching, Structured clinical debriefing, and/or Supportive supervision.

8. References and Reporting Standards

a) Ensure all references are complete and formatted according to PLOS ONE guidelines. Some currently lack volume, issue, or page numbers.

b) Submit a completed COREQ checklist as a supplementary file to demonstrate adherence to qualitative reporting standards.

**Do you want your identity to be public for this peer review?** For information about this choice, including consent withdrawal, please see our Privacy Policy

Reviewer #1: No

Reviewer #2: **Yes: ** Yong-chuan Chen

---

## [Author Response · Author response to Decision Letter 1]

27 Aug 2025

August 21, 2025

Dawit Getachew Gebeyehu, MPH

Academic Editor

PLOS ONE

Dear Editor,

Thank you for your further consideration of our manuscript titled “Challenges encountered by midwives performing basic neonatal resuscitation in health facilities in Kinshasa, Democratic Republic of the Congo” (PONE-D-24-56077). We thank you and the reviewers for their helpful comments, and respectfully submit the final revised manuscript. Below is a point-by-point response to the reviewers’ comments.

Sincerely,

Jackie K. Patterson, MD, MPH

Associate Professor of Pediatrics with Tenure

Division of Neonatal-Perinatal Medicine

University of North Carolina at Chapel Hill

Response to Reviewers

Reviewer #1:

Thank you for the opportunity to review your manuscript. Your study presents a well-structured qualitative analysis of the challenges midwives face in implementing neonatal resuscitation in the Democratic Republic of the Congo (DRC). It effectively highlights key barriers, including human resource limitations, inadequate equipment, stress, and emotional burden. Below are some comments that I hope will be helpful in refining your manuscript for publication.

Introduction

1) The introduction outlines neonatal mortality statistics and the importance of timely resuscitation, but it does not explicitly state the gap in existing knowledge that this study seeks to fill.

Response:

We added the following sentence explicitly stating the gap this study seeks to fill:

“A decline in knowledge and skills following training has been clearly identified as a prominent barrier to quality newborn resuscitation. (13, 14) However, little is known about the other barriers that contribute to poor performance of neonatal resuscitation such as emotional experiences and their influence on bedside decision-making.” (Introduction, lines 104-108)

2) Please provide the reader with further details about basic resuscitation in DRC.

Response:

We added the following sentence to provide further details about basic resuscitation in DRC:

“Specifically in the DRC, we have demonstrated that even after training in resuscitation, providers are delayed in initiating ventilation (median time to ventilation of 347 seconds after birth), and ventilation is frequently interrupted with stimulation and suctioning.” (13, 14) (Introduction, lines 102-104)

Material and Methods

3) The context in which the study is taking place is missing. Please include a paragraph about your setting in which the study takes place. What care level where the facilities at? Number of births? Neonatal deaths, Resuscitations, etc.

Response:

We added the following paragraph in a new subsection entitled “Setting:”

“This study took place in three Catholic health facilities situated in three different health zones in Kinshasa, the capital city of the DRC. The approximate number of annual births at these facilities ranged from 1,300 to 3,900. Approximately 3% of neonates received bag-mask ventilation at birth at these facilities, and the neonatal mortality rate was approximately 13 per 1000 live births. All of the facilities practiced basic resuscitation; none of the three facilities used electronic fetal heart monitoring during labor, and none offered Cesarean section.” (Materials and Methods, Setting, lines 125-130)

4) The description of the focus group discussions (FGDs) is detailed, but some additional clarifications would improve transparency. Were there any efforts to minimize groupthink bias in FGDs?

Response:

We added the following sentence describing efforts to minimize groupthink bias:

“To minimize groupthink bias, the facilitators were neutral in their responses, encouraged individual opinions, and discouraged side conversations among participants.” (Materials and Methods, Study design and procedures, lines 160-162)

5) Did the facilitators have prior relationships with participants that could have influenced responses?

Response:

We added the following sentence regarding prior relationships between the facilitators and the participants:

“While EM did not have a prior relationship with the participants, many of the participants knew DI as he had recently coordinated a clinical trial on newborn resuscitation in those sites.” (Materials and Methods, Study design and procedures, lines 149-151)

6) As a reader, I am not entirely clear on the type of analysis you have conducted. The analysis procedure refers to both content and thematic analysis, making it difficult to understand your analytical process and how your themes emerged. For example, you reference Elo and Kyngäs, but based on my understanding, your analysis does not align with theirs. Please revisit your references and provide a clearer explanation of how your content analysis was conducted. This is particularly important since you have counted your themes, which is not mentioned in the analysis section.

Response:

We have clarified our analysis procedures with the following text:

“Using Atlas-ti 9 (Scientific Software development, Berlin), we analyzed transcripts following an inductive content approach in order to identify emergent themes and trends in the data.(16-18) First, French transcripts were read and re-read by two research team members to familiarize themselves with the content of the transcripts. For this step, we used a voice-centered relational method with reflexive elements built in that included reading the interview text three or more times. One of the readings involved using a worksheet technique to lay out the respondent’s words in one column and react to and interpret the text in an adjacent column.(19) In this ‘reader-response’ technique, the researcher puts himself, his background, history and experiences in relation to the respondent, reading the narrative on his own terms and listening for how he is responding emotionally and intellectually to this person. This allows researchers to examine how and where some of their assumptions and views might affect their interpretation of the respondent’s words, or how they later write about the person. For example, since both facilitators were physicians with prior experience providing newborn resuscitation, they were aware of the tendency for midwives to suction all babies, including those who were breathing. This prior experience influenced the way they coded quotes related to suctioning, ultimately assigning them a category of ‘tendency to stick to the routine.’ In a second step, both readers independently developed a coding sheet, assigning codes that emerged from reading. With each subsequent FGD, they assessed consistency in the data, meaning saturation and code saturation to determine if thematic saturation had been reached. In a third step, the two readers then discussed their coding sheets and together created sub-categories and categories by consensus. Subsequently, we labeled participant individual comments, counted them to provide figures and orders, and then organized them into labels, sub-categories, and categories (in ascending order). The team used notes from the meetings to corroborate their categorization schema (20) and refined as needed. While participants were not approached to provide feedback on the findings, analytical outputs were refined through discussion with other research team members from different backgrounds and cultures.” (Materials and Methods, Data processing and analysis, lines 178-212)

Results

7) The manuscript often states the number of midwives mentioning a particular theme (e.g., "n=6"), but this is inconsistent. Some sections include this, while others do not. Consider whether the quantification is required, and if so, be consistent.

Response:

We added quantification where previously missing (see tracked changes in Results section).

8) The results would benefit from a more in-depth analysis.

Response:

We added text exploring underlying mechanisms and/or opportunities to mitigate challenges for the following subthemes: insufficient monitoring during labor (lines 278-285), poor systems to support maintenance of skills (lines 293-312), infrequent opportunity to resuscitate (lines 315-323), tendency to stick to the routine (lines 379-390), acute stress during resuscitation (lines 404-413), and moral distress after unsuccessful outcome (lines 416-424).

Discussion

9) The discussion relates findings to previous research, but it lacks a discussion in relation to research on midwives in DRC (simulation, education, work environment).

Response:

Given other requested additions to the discussion, we elected to add more detail regarding research on midwives in DRC to the introduction:

“Although healthcare provided by midwives is a core strategy for improving newborn health in low-resource settings, the midwifery profession in the DRC is challenged by poor quality pre-service training, unsupportive organizational systems and inadequate pre-conditions in the work environment.(4, 5) Despites these challenges, Congolese midwives view their profession as a calling and love their work, motivating them to continue in their workplace despite the difficult work environment and low professional status. (4) Enhancing the quality of care delivered by midwives in the DRC is critical to improving maternal and newborn health outcomes.” (Introduction, lines 72-79)

10) The limitations section is well written but could be expanded to discuss: possible social desirability bias in FGDs and the generalizability of findings beyond Kinshasa, given urban-rural disparities in healthcare resources.

Response:

We added social desirability bias and the influence of urban-rural disparities on the generalizability of our findings as additional limitations. The section on the influence of urban-rural disparities reads as follows:

“Although our sample was drawn from high-performing urban facilities, we believe the challenges identified—particularly the emotional burden of resuscitation—are likely transferable to similar low-resource maternity settings. Further research is needed to explore these dynamics in rural or under-resourced contexts.” (Discussion, lines 568-572)

11) Consider adding the following references:

a. Berg et al (2022) Implementation of a three-pillar training intervention to improve maternal and neonatal healthcare in the Democratic Republic Of Congo: a process evaluation study in an urban health zone

b. Bogren et al (2020) Midwives’ challenges and factors that motivate them to remain in their workplace in the Democratic Republic of Congo—an interview study

c. Bogren et al (2021) Barriers to delivering quality midwifery education programmes in the Democratic Republic of Congo — An interview study with educators and clinical preceptors – ScienceDirect

d. Baba et al (2020) 'Being a midwife is being prepared to help women in very difficult conditions': midwives' experiences of working in the rural and fragile settings of Ituri Province, Democratic Republic of Congo – PubMed

Response:

We have added three of the above references to the manuscript (see both the introduction and discussion sections).

Reviewer #2:

This manuscript addresses a critically important issue in global maternal and newborn health by exploring the challenges encountered by midwives in implementing basic neonatal resuscitation in the Democratic Republic of the Congo (DRC). The qualitative design is appropriate for the research question, and the study provides rich, meaningful insights into both structural and psychological barriers to quality neonatal care in low-resource settings. It is a valuable contribution to the field and presents data that are largely consistent with the conclusions drawn. However, several key areas require attention and improvement before the manuscript can meet the methodological and editorial standards expected by PLOS ONE. My detailed comments are as follows:

Abstract

1) Please clarify in the abstract that this is a qualitative study employing focus group discussions and inductive content analysis. At present, the methodology is described in general terms, which may hinder readers from immediately identifying the study design. I recommend adding the phrase: "We conducted a qualitative study using focus group discussions and inductive content analysis..."

Response:

We added the phrase you suggested to the abstract (see abstract, lines 32-33).

Introduction

2) Consider explicitly articulating the gap in the literature, particularly concerning midwives’ emotional experiences and bedside decision-making. This would position your study more clearly as a novel contribution. You may also cite relevant literature on emotional labor or the psychological burden associated with resuscitation to enhance theoretical grounding.

Response:

We added the following sentence explicitly stating the gap this study seeks to fill:

“While a decline in knowledge and skills following training has been clearly identified as a prominent barrier to quality newborn resuscitation in the literature. (13, 14) However, little is known about the other barriers that contribute to poor performance of neonatal resuscitation such as emotional experiences and their influence on bedside decision-making.” (Introduction, lines 104-108)

We also cited relevant literature on stress during neonatal resuscitation to enhance theoretical grounding:

“These barriers may have a prominent influence on performance given prior research indicating increased stress in healthcare workers during neonatal resuscitation. (15) Furthermore, cumulative number of losses experienced is a predictor of stress, and thus stress during neonatal resuscitation in low-resource settings may be particularly high given the relatively high incidence of adverse outcomes.” (15) (Introduction, lines 108-112)

3) While the rationale for examining implementation barriers is sound, the justification for using a qualitative design should be expanded. Explain why qualitative methods were necessary to capture midwives’ experiences, and how they provide unique insights beyond what quantitative studies offer.

Response:

We added the following statement justifying our selection of qualitative methods for this study:

“Investigation of such barriers requires qualitative methods to illuminate the lived experiences of midwives and explore the complexity of their decision-making during urgent situations.” (Introduction, lines 112-119)

Materials and Methods

4) Expand on interviewer characteristics (e.g., gender, professional background, training) and any steps taken to minimize interviewer bias. These are key elements of the COREQ checklist and contribute to the transparency and trustworthiness of the study.

Response:

We expanded on facilitator characteristics with the following addition:

“Two male team members (EM and DI) with prior qualitative data collection experience conducted FGDs in February 2021. EM has an MD, MPH and a PhD in global health and was an associate professor at the Kinshasa School of Public Health; DI has an MD and was a PhD student in maternal and child health. While EM did not have a prior relationship with the participants, many of the participants knew DI as he had recently coordinated a clinical trial on newborn resuscitation in those sites.” (Methods, Study design and procedures, lines 146-151)

We also described how we minimized facilitator bias:

“To minimize facilitator bias, the facilitators used an FGD guide to structure the discussion, engaged in active listening, and asked open-ended questions.” (Methods, Study design and procedures, lines 152-154)

5) Clarify how translation fidelity was ensured in the process from Lingala to French and then English. This is essential for data trustworthiness. Please state whether transcripts were back-translated or reviewed by bilingual team members to validate accuracy.

Response:

We added the following statement:

“Transcripts were back-translated and reviewed by bilingual team members to validate accuracy.” (Materials and Methods, Data processing and analysis, lines 177-178)

Materials and Methods: Data Analysis

6) Provide specific examples of how reflexivity and the worksheet technique influenced coding or thematic interpretation. Currently, the reflexivity component appears superficial.

Response:

We provided the following specif

---

## [Decision Letter · Decision Letter 1]

2 Oct 2025

Challenges encountered by midwives performing basic neonatal resuscitation in health facilities in Kinshasa, Democratic Republic of the Congo

PONE-D-24-56077R1

Dear Dr. Patterson,

We’re pleased to inform you that your manuscript has been judged scientifically suitable for publication and will be formally accepted for publication once it meets all outstanding technical requirements.

Kind regards,

Dawit Getachew Gebeyehu, MPH

Academic Editor

PLOS ONE

Additional Editor Comments (optional):

Reviewers' comments:

Reviewer's Responses to Questions

**Comments to the Author**

Reviewer #1: All comments have been addressed

Reviewer #2: All comments have been addressed

2. Is the manuscript technically sound, and do the data support the conclusions?

Reviewer #1: Yes

Reviewer #2: Yes

3. Has the statistical analysis been performed appropriately and rigorously?

Reviewer #1: N/A

Reviewer #2: N/A

4. Have the authors made all data underlying the findings in their manuscript fully available?

Reviewer #1: Yes

Reviewer #2: Yes

5. Is the manuscript presented in an intelligible fashion and written in standard English?

Reviewer #1: Yes

Reviewer #2: Yes

Reviewer #1: (No Response)

Reviewer #2: Thank you for your thoughtful and comprehensive revisions to this manuscript. After carefully reviewing both the authors’ point-by-point responses and the revised text, I am satisfied that all previously raised concerns have been adequately addressed. The methodology is now presented with much greater clarity, including detailed descriptions of the study setting, participant recruitment, facilitator characteristics, translation fidelity, and analytic procedures. The addition of reflexivity and a clear explanation of coding and consensus processes substantially enhances methodological rigor.

The results section has been reorganized into coherent themes and subthemes, supported by a thematic table and consistent quotation labeling. This restructuring improves both readability and analytic depth. The discussion has been expanded and enriched by the integration of Cognitive Load Theory and attention to organizational culture, supervision, and debriefing strategies, which strengthen the interpretive framing of the findings. Importantly, the limitations section now explicitly addresses the potential for social desirability bias and the limited transferability of findings beyond the urban context of Kinshasa.

The manuscript also meets PLOS ONE reporting requirements, with submission of the COREQ checklist and full data availability through a public repository. The text is written in clear and intelligible English, and the revisions have improved overall flow and presentation.

In sum, the revised manuscript is methodologically sound, clearly presented, and makes a valuable contribution to the literature on the challenges midwives face in implementing neonatal resuscitation in low-resource settings. I have no further substantive concerns, and I believe the work is now ready for publication.

**Do you want your identity to be public for this peer review?** For information about this choice, including consent withdrawal, please see our Privacy Policy

Reviewer #1: No

Reviewer #2: **Yes: ** Yong-Chuan Chen

---

## [Editor Report · Acceptance letter]

PONE-D-24-56077R1

PLOS ONE

Dear Dr. Patterson,

I'm pleased to inform you that your manuscript has been deemed suitable for publication in PLOS ONE. Congratulations! Your manuscript is now being handed over to our production team.

Kind regards,

on behalf of

Mr. Dawit Getachew Gebeyehu

Academic Editor

PLOS ONE